# Now I Remember!
# Episodic Memory For Reinforcement Learning

## Abstract

Humans rely on episodic memory constantly, in remembering the name of someone they met 10 minutes ago, the plot of a movie as it unfolds, or where they parked the car. Endowing reinforcement learning agents with episodic memory is a key step on the path toward replicating human-like general intelligence. We analyze why standard RL agents lack episodic memory today, and why existing RL tasks do not require it. We design a new form of external memory called Masked Experience Memory, or MEM, modeled after key features of human episodic memory. To evaluate episodic memory we define an RL task based on the common children's game of Concentration. We find that a MEM RL agent leverages episodic memory effectively to master Concentration, unlike the baseline agents we tested.

## 1 Introduction

From a neurobiological perspective, *episodic memory* is a key component of human life — remembering the name of a new acquaintance, recalling the plot of a movie as it unfolds, or realizing where the car is parked, are all examples of how we use episodic memory[1] to store and recall novel information. If a person's ability to form and retrieve new episodic memories is lost, as in advanced Alzheimer's disease, the person is severely incapacitated as a result. Although today's standard Reinforcement Learning (RL) agents possess forms of procedural and semantic memory (Gershman & Daw, 2017), they lack any functional equivalent of episodic memory. Our motivation is to expand the general intelligence of RL agents by imbuing them with a useful form of episodic memory.

Human episodic memories appear to be records of experience that are re-experienced when associatively recalled (Conway, 2009). In RL, fundamental experiences are termed *observations*. Accordingly, we propose the following working definition: *Episodic memory for an RL agent is the ability to leverage details of a past observation that is similar to the current observation*. This definition implies that an agent would exercise episodic memory by doing certain things at specific points in time, including

1. At the time of the old observation, the details of that observation must be stored somewhere in the agent. This stored record is the episodic memory.
2. Later, when another observation arrives, it must somehow be compared with the stored observations. If one of those is sufficiently similar, then the details of the old observation must be retrieved from memory. There are different implementations of similarity and retrieval. We will propose a concrete one later.
3. After retrieving the details of the old observation that is similar to the new one, the agent must be able to utilize that information to benefit it's pursuit of reward.

Designing an RL agent with episodic memory is one challenge, and designing an RL *task* to evaluate episodic memory in an agent is another. The main difficulty is that unless the task is very carefully designed, the RL agent may find a way to solve the task using other learning abilities besides episodic memory. To illustrate, we briefly introduce the RL task that we will present later in detail.

To evaluate an agent's episodic memory ability, we introduce the Concentration task based on the card game of the same name. Concentration is a memory game with the goal of identifying matching

---

[1]Throughout, *episodic memory* refers to human episodic memory, and is not to be confused with episodes in a Markov decision process Sutton & Barto (1998).

pairs of cards among a large set of face-down cards. During play, one card at a time is temporarily revealed to the player who must correctly memorize and recall the locations of each pair. Concentration tests episodic memory by requiring an agent to leverage past observations of cards and their locations in order to succeed. In our variant of Concentration, cards are not limited to the standard deck and are instead randomly generated for each game, so each card pair is unique and never before seen in the agent's lifetime. Unique cards test the agent's ability to use episodic memory to reason about the identities and locations of the cards that are seen within the current episode, rather than learning to recognize specific cards.

Recently, the capabilities of intelligent agents have greatly expanded through the combination of deep learning and reinforcement learning. Deep RL agents have achieved notable success outperforming humans on Atari games (Mnih et al., 2015). However, many of the hardest tasks in which RL agents still fail to surpass humans are fraught with the difficulties of sparse rewards, partial observability, and a limited amount of samples. Equipping an RL agent with memory is a promising approach to tackling some of these challenges, and has attracted a growing amount of interest in the research community.

Recurrent neural networks such as LSTMs are commonly used as controllers (Hausknecht & Stone, 2015; Mnih et al., 2016). LSTMs can be trained to maintain and use information on timescales of tens of steps, but have trouble learning over longer sequences. Additionally, LSTMs do not store observations as discrete entities, so it is unclear how an LSTM could compare a never-before-seen observation (such as a unique card) with detailed instances of past observations, which also may have occurred only once.

Memory augmented neural networks provide storage capabilities beyond those of an LSTM. One such architecture, the differentiable neural computer (DNC) (Graves et al., 2016) has been shown to be capable of handling several different memory-based tasks. We evaluate the DNC on Concentration, but discover that it has difficulty reusing elements of its memory matrix.

The key contributions of this paper are:

- We propose a working definition of episodic memory for RL agents.
- We introduce the Concentration task for evaluating episodic memory.
- We present the Masked Experience Memory (MEM) architecture, a new type of external memory designed to provide an RL agent with human-inspired episodic memory, and incorporating a novel improvement over cosine similarity for content-based addressing.
- We empirically demonstrate that MEM successfully enables an RL agent to solve the Concentration task by remembering the identities and locations of cards it has seen only once.
- We show that baseline RL agents (LSTM-based and DNC-based) fail to solve the task.

## 2  RELATED WORK

Neither neural network weights nor activation states support the storage and associative retrieval of discrete, one-shot experiences necessary for episodic memory. Neural network weights change too slowly to store samples as individual experiences. Fast weights (Ba et al., 2016) are one approach to storing information from discrete samples in weights, but we find no published evaluations of fast weights on episodic memory tasks.

Many external memory architectures have been proposed for augmenting the capabilities of neural networks in the supervised learning setting. Some of these were evaluated on data samples that occur a relatively small number of times (Vinyals et al., 2016; Pickett et al., 2016; Santoro et al., 2017; Kaiser et al., 2017). Adapting these architectures to RL tasks is non-trivial. For instance, the memory module of Kaiser et al. (2017) requires ground truth output labels to create and modify memories.

A few augmented memory architectures have been applied to the more challenging setting of deep reinforcement learning (Zaremba & Sutskever, 2015; Oh et al., 2016; Blundell et al., 2016; Graves et al., 2016; Pritzel et al., 2017). These external memory schemes were shown to improve learning on certain tasks, typically by increasing sample efficiency. But to our knowledge none of them were evaluated on episodic memory tasks.

Despite the growing body of literature on active memory, most environments do not capture the diversity of observations required to test episodic memory. For instance, in maze tasks such as the the T-Maze studied by Oh et al. (2016), the agent must remember a color seen at the start of the episode, then use that information later to move in the correct direction at the T-junction. But since only two colors are ever displayed at the start of the maze, the agent can learn to associate the floor tile color with the correct actions using neural network weights. In contrast, an episodic memory task like Concentration presents many previously unseen observations which must be handled correctly without prior exposure.

With the differentiable neural computer (DNC), Graves et al. (2016) showed than an RL agent could use a memory matrix to buffer and use data to complete a moving blocks puzzle (Mini-SHRDLU). We show that Mini-SHRDLU can be decomposed into separate data buffering and problem-solving subtasks (Appendix B), neither of which requires human-like episodic memory.

Episodic memory tasks can be viewed as a special type of transfer learning. Transfer and multitask learning (Taylor & Stone, 2009) involve evaluating the agent on a novel task (e.g. with new observations, rewards, transition dynamics). Episodic memory tasks such as Concentration feature an endless stream of novel observations but unchanged rewards and dynamics. Prior work on transfer learning has relied on techniques from DeepRL and model compression (Parisotto et al., 2016; Rusu et al., 2016; Devin et al., 2017).

Concentration can be viewed as an episodic one-shot learning task in which novel observations must be correctly memorized and recalled after a single viewing. Prior work on few-shot image classification (Vinyals et al., 2016; Rezende et al., 2016) has used learned metric spaces and siamese networks (Gregory Koch, 2015).

## 3 MASKED EXPERIENCE MEMORY (MEM)

The Masked Experience Memory (MEM) architecture imbues an RL agent with the ability to leverage details of a past observation that is similar to the current observation. MEM's focus on observations differentiates it from DNC, which writes vectors to memory which are abstract in the sense that they are not bound to agent observations. Similarly, while MEM's read operation compares past observations with the current observation, DNC's read operation compares previously written memory vectors to an abstract read vector having no necessary connection to any observation. In these respects, DNC's memory mechanism is strictly more general than that of MEM, possessing more freedom of representation, as well as more potential challenges in training. This makes DNC a valuable baseline for comparison to MEM.

Each MEM memory write operation copies the last observation into a fixed-size memory store, while the oldest memory is dropped from the store. Other external memory implementations share this general method of writing memories (Sukhbaatar et al., 2015; Oh et al., 2016). This corresponds to the rapid forgetting of human episodic memories (Conway, 2009). Despite its simplicity, we view this design as applying the useful prior assumption that the most recent history is often the most relevant to selecting a good next action.

Each memory read operation compares the current observation with all past observations in memory, and returns a vector calculated as a weighted sum of all memories. This part of the read operation is the same as that used by most content-based addressing memory architectures, given by:

$$R = \sum_{i=1}^{N} b_i M_i \,, \qquad\qquad b_i = \frac{\exp(Q_i)}{\sum_{j=1}^{N} \exp(Q_j)} \,,$$

where the memory matrix $M$ contains $N$ memories, each implemented as a real-valued vector of $D$ dimensions (elements); the read vector $R$ (also of length $D$) is a weighted average of all memories; the read weighting vector $b$ is a normalized probability distribution over memories; finally, each memory's weight $b_i$ is a function of that memory's similarity ($Q_i$) to the current read key vector.

There are various choices for the similarity function $Q$, such as the popular cosine function. Here, we propose to use vector quadrance scaled by an explicit mask over vector elements:

$$Q_i = -\exp(z) \sum_{d=1}^{D} a_d (s_d - M_{id})^2 \,, \qquad\qquad a_d = \frac{\exp(w_d)}{\sum_{k=1}^{D} \exp(w_k)} \,,$$

where each memory's similarity ($Q_i$) to the current read key (state vector $s$) is the squared Euclidean distance (quadrance) between them scaled by the corresponding element of the mask vector $a$; the mask vector $a$ is a learned distribution over memory dimensions; the mask weight vector $w$ and attention distribution sharpness parameter $z$ are trained by gradient descent.

The mask weight vector $w$ is intended to learn which memory dimensions should be used as the lookup key for memory read operations, which in turn determines the attention distribution over memories. For instance, as a particular mask weight $w_i$ increases, the corresponding mask element $a_i$ will also increase, causing that element of each memory to contribute more to all the similarity calculations.

MEM's usage of a mask vector in calculating vector similarity is designed to avoid a potential source of noise associated with a commonly-used similarity calculation, cosine similarity: $D(u, v) = \frac{u \cdot v}{|u||v|}$

|  |  |  |  |  |  | Dot prod | Mag | Cos Sim |
|---|---|---|---|---|---|---|---|---|
| **Key vector (u)** | 1 | -1 | 0 | 0 | 0 |  | 1.414 |  |
| **Memory vector 1 (v)** | 1 | -1 | 1 | -1 | 1 | **2.0** | 2.236 | 0.632 |
| **Memory vector 2 (v)** | 0 | -1 | 0 | 0 | 0 | 1.0 | 1.000 | **0.707** |

Figure 1: **Pathological Example of Cosine Similarity**: Although memory vector 1 is identical to the completed (non-zero) portion of the key vector, cosine similarity judges memory vector 2 to be more similar to the key vector.

Since cosine similarity measures the angular similarity between two vectors by normalizing out their magnitudes, it is ideally suited for comparing word count vectors from documents of different lengths, for instance. But in the general case of content-based addressing, it is often intuitive to view the read key as partially specified, with zeros in the unspecified elements. From that perspective the read operation replaces the zeros with values from a memory or memories that best match the read key. This avoids the complexity of separate key and value vectors.

However, applying cosine similarity in this way can add noise to the similarity calculation, as illustrated in Figure 1. Since cosine similarity normalizes the dot product by the magnitudes of both vectors being compared, the supposedly masked-out elements of the memory vector can still affect the results. This noise becomes large as the non-zero portion of the key vector becomes small.

MEM avoids this problem by using an explicit mask to select which vector elements will participate in the similarity measurement.

# 4 RL AGENT ARCHITECTURES

In this section, we discuss how to use MEM in an RL agent. An RL agent aims to maximize its expected long-term return by acting in an initially unknown environment (Sutton & Barto, 1998). In each step, it makes an observation about the environment, takes an action, then receives an immediate reward and next observation. While there exist many algorithms in the literature, for concreteness, we use one of the most effective algorithms known as Asynchronous Advantage Actor-Critic (A3C) (Mnih et al., 2016) to explain how to incorporate MEM, and run experiments with this instantiation. The use of our memory architecture is similar for other RL algorithms.

Since the environment is only partially observable in many real-world problems as well as in the game of Concentration, the observations received in individual time steps are not Markovian (Sutton & Barto, 1998). Information collected from past observations should therefore be remembered and used to make a decision at each step. One possibility is to use an LSTM to compress past observations into a fixed-length vector, which is used to approximate a Markovian state of the environment. This approach makes use of a limited form of memory, and is illustrated in Figure 2 (left panel) where LSTM is used inside both the actor and critic networks of A3C.

The more general DNC-based agent uses separate actor and critic DNCs, each containing its own LSTM controller and memory matrix, as shown in the right panel of Figure 2. Note that the same approach is taken by Graves et al. (2016) for the Mini-SHRDLU task.

Our proposed architecture, based on MEM, is given in the middle panel of Figure 2. It uses separate actor and critic LSTM controllers which share the same memory store. For episodic RL tasks, we clear MEM's memory store at the beginning of each episode, although other possibilities exist.

For all agents on every time step, each LSTM controller receives as inputs the current observation vector from the environment, concatenated with a one-hot vector representing the last action taken, plus the reward just received. Each LSTM also receives as input the most recent output from the memory store (whether MEM or DNC). In the case of MEM, a memory similarity strength value is also passed as an additional feature to the LSTMs. In the case of DNC, the memory store's output is immediately concatenated with the output from its LSTM running to the output layer of its network.

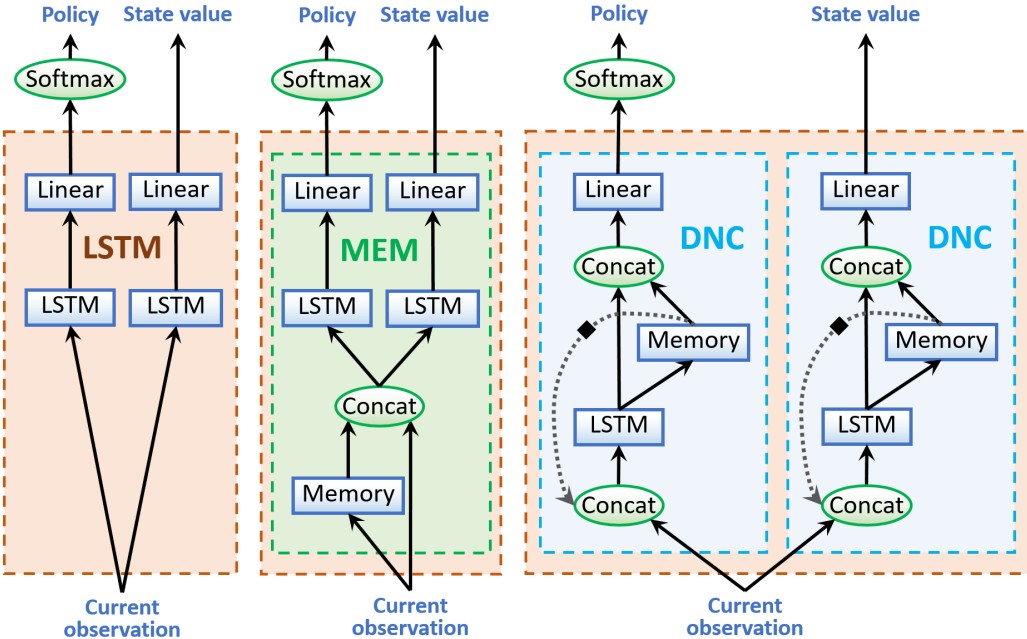

Figure 2: The three RL agent architectures evaluated on Concentration.

## 5  CONCENTRATION TASK

To the best of our knowledge, no existing RL benchmark task could unambiguously evaluate episodic memory in RL agents. We therefore designed a new task for this purpose, derived from the common children's memory game of Concentration, as described in Wikipedia.[2] The game is played with a deck of cards in which each card face appears twice. At the start of each game, the cards are arranged face down on a flat surface. A player's turn consists of turning over any two of the cards. If their faces are found to match, the player wins those two cards and removes them from the table, then plays again. If the two cards do not match, the player turns them face down again, then play passes to the next player. The game proceeds until all cards have been matched and removed from the table. The winning strategy is to remember the locations of the cards as their faces are revealed, then use those memories to find matching pairs.

We convert the Concentration game into a single-player, episodic RL task. The agent occupies one cell at a time within a square grid of cells, each of which may be empty or may contain one card. The grid is just large enough to hold all the cards. Each card may be either face up or face down on any given time step. The agent's available actions are to take one step in any of the four directions, or to flip over the card at the current location. Whenever two cards fail to match, they automatically turn face down on the next time step. Whenever two cards match, the agent is rewarded, and the two cards are automatically removed from the grid on the next time step. The episode terminates when the last two cards are matched and removed, or when enough time steps have passed for all cards to have been removed. The agent receives a small penalty for each card it flips over.

---

[2]URL: https://en.wikipedia.org/wiki/Concentration_(game)

The major issue in designing this as a test of episodic memory is how to represent the agent's observation of a card face. The simplest scheme would be to represent each card face by a one-hot vector of length $N/2$, where $N$ is the number of cards in the deck. But this would allow the agent to solve the task without relying on episodic memory, because the total number of card faces and card positions could be small enough for the agent's network (over many training episodes) to dedicate a different unit to every possible card-plus-position combination. Then in the course of play, whenever a card was revealed, the network would only need to toggle the activation state of the unit representing that particular card and position. If two units corresponding to a matching pair were both active, the agent would know their locations and could then proceed to flip both of them over.

Instead of using one-hot vectors, each card face could be represented by a complex image, such as an Omniglot character, to be processed by a convolutional neural network. But the network could still employ the strategy described above, only at a higher embedding level in the network, after learning the fixed identities of the cards.

So to make this an unambiguous test of episodic memory, we generate new images for all card faces at the start of each episode. This is equivalent to playing just one game of Concentration with a deck of cards, then replacing it with a new deck of cards having different face images for the next game, etc. This ensures that each image appears in no more than one episode, making it impractical for an agent's neural network to learn each image as a persistent entity from game to game.

Instead of using images composed of pixels, we define each card face to be a random real-valued vector of length 6. This can be thought of as an image embedding vector learned at some upper level of a CNN. We did not try any other image sizes. Each card face image (to appear on two cards) is generated by randomly selecting six real numbers in the range $[0, 1]$. If the resulting vector is too close to an already generated vector, based on a fixed Euclidean distance threshold, the vector is randomly regenerated.

The agent's observation vector contains six components:

1. A one-hot state vector for the cell occupied by the agent plus each of the 8 surrounding cells. Each cell has 4 possible states: card with face down, card with face up, no card, and off-grid.
2. A 6D real vector for the card at the agent's current location. This vector contains zeros if the cell does not contain a face-up card.
3. A one-hot vector identifying the row currently occupied by the agent.
4. A one-hot vector identifying the column currently occupied by the agent.
5. A one-hot vector reporting the previous action taken by the agent.
6. A single real number equal to the reward just received by the agent.

The agent's performance is evaluated in terms of card-pair matches per card flip, which is closely tied to the reward received. Agents are not directly penalized for spending time wandering around the grid, but reward per time step is maximized by clearing the board quickly. For the experiments reported here, all tasks used 8 cards on a $3 \times 3$ grid.

## 6 EXPERIMENTAL RESULTS

We tested the following six agents on the Concentration task:

1. MEM agent we proposed in Section 3.
2. DNC agent of Graves et al. (2016)
3. TensorFlow LSTM agent (lacking external memory).
4. Sonnet LSTM agent (obtained by disabling DNC's memory matrix).
5. Colorblind MEM agent (control).
6. Colorblind TensorFlow LSTM agent (control).

We chose DNC as a baseline because of its particularly powerful form of external memory, and because it differs from our own architecture in several respects. The colorblind agents serve as controls, being unable to distinguish card faces. All agents were trained using asynchronous advantage actor-critic (A3C) (Mnih et al., 2016) and differ only in their memory architectures. After finding each agent's optimal hyper-parameter settings through extensive search, we trained and evaluated each

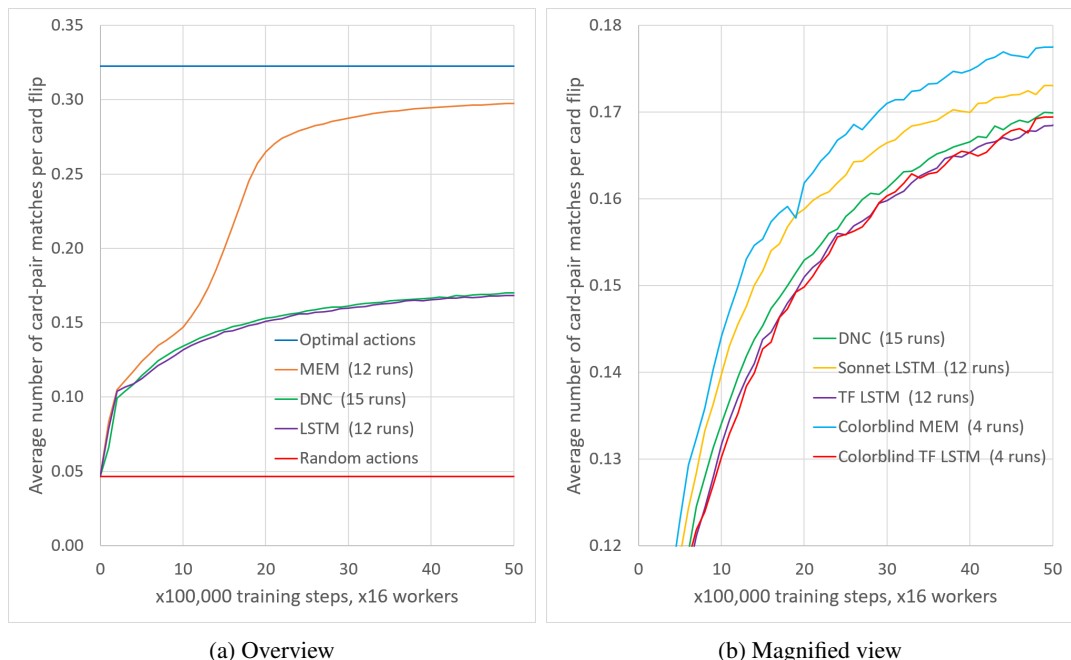

(a) Overview                                                    (b) Magnified view

Figure 3: **The MEM agent effectively utilizes episodic memory to solve the Concentration task:** During training, counters keep track of the number of cards flipped by all 16 worker agents, as well as the number of card-pair matches obtained by the agents. After every $100,000$ training steps these counts are used to calculate the matches per flip for that period, then reset to zero. This produces a trailing estimate (including exploratory actions) of the agent's performance on the task.

Table 1: Hyper-parameter settings

|  | MEM | TF-LSTM | DNC | So-LSTM | Parameter settings tested |
|---|---|---|---|---|---|
| **Tuned parameters** |  |  |  |  |  |
| Learning rate | 0.0001 | 0.0001 | 0.0001 | 0.0001 | 0.00001, 0.00001, 0.00005, 0.0001, 0.0002, 0.0005 |
| Optimizer | Adam | Adam | RMSProp | Adam | Adam, RMSProp |
| Discount factor | 0.99 | 0.99 | 0.99 | 0.99 | 0.60, 0.65, 0.70, 0.75, 0.80, 0.90, 0.99 |
| MEM init quadrance scale | 50 |  |  |  | 20, 30, 40, 50, 60 |
| DNC read heads |  |  | 2 |  | 1, 2, 4 |
|  |  |  |  |  |  |
| **Fixed parameters** |  |  |  |  |  |
| Memory size | 50 |  | 2x50 |  |  |
| Memory word size | 54 |  | 60 |  |  |
| LSTM size | 2x128 | 2x128 | 2x128 | 2x128 |  |
| Training window length | 32 | 32 | 32 | 32 |  |
| Gradient clip threshold | 50 | 50 | 50 | 50 |  |
| Per-match reward | 25 | 25 | 25 | 25 |  |
| Flip penalty | -0.005 | -0.005 | -0.005 | -0.005 |  |
| Asynchronous workers | 16 | 16 | 16 | 16 |  |
| Entropy cost coefficient | 0.01 | 0.01 | 0.01 | 0.01 |  |

agent using its optimal settings on a number of additional runs. Table 1 gives the hyper-parameter settings used on Concentration, both tuned and fixed. Key results are collected in Figure 3a.

## 7 DISCUSSION

The optimal mean performance attainable by an agent with perfect episodic memory is shown at the top of Figure 3a (Velleman & Warrington, 2013). Only the MEM agent learned a near-optimal policy. The baseline LSTM-A3C agent's results were overlapped with those of its colorblind version

3b, demonstrating that the LSTM-A3C agent never learned to remember the locations of the cards it saw. The Sonnet LSTM agent performed consistently better than the TensorFlow LSTM agent 3b, though not by a large amount. Both implementations claim to be based on Zaremba et al. (2014), so the difference in behavior is unexpected.

Despite being unable to see the card faces, the colorblind MEM agent 3b still performed a bit better than any of the LSTM agents, indicating that it found some other strategy (not based on card faces) to derive a small amount of gain from its external memory.

Even after dozens of trial settings over a wide range of hyper-parameters, the DNC agent performed only very slightly better than the LSTM-A3C agent, and noticeably worse than its own recurrent controller alone, the Sonnet LSTM agent. We did not attempt curriculum learning. Appendix A presents a detailed investigation into the causes of DNC's poor performance on this type of task.

Performing ablation studies on the MEM architecture, we found that using the mask (instead of cosine similarity) and Euclidean distance squared were both essential to scoring above the LSTM-A3C baseline. Adaptation of the sharpness term turned out to be essential for stable results. On the other hand, the similarity strength feature provided no measurable benefit.

As intended, MEM's most positive learned mask weights were the ones for the six card face dimensions. At convergence of the best MEM model, $83\%$ of the mask's mass was concentrated on those six elements, even though they constitute only $11\%$ of the observation vector's $54$ elements.

## 8 Conclusions

We have defined episodic memory for RL agents, provided an unambiguous test for evaluating it, and presented an implementation of episodic memory that corrects a problem with current content-based addressing methods. Our results show that this MEM architecture, designed to emulate specific aspects of human episodic memory, is able to use that memory effectively in the Concentration task by remembering the locations of cards it has seen only once before. This is in sharp contrast to the other agents tested, which never learned to remember card locations. The code to replicate this work will be made public prior to the conference.

MEM represents the initial step on a path towards more robust and powerful episodic memory for RL agents. We plan to extend MEM in several significant ways:

1. Making the mask weights context-sensitive so that read key vectors can quickly shift to cover different aspects of experience depending on the situation.
2. Expanding the memory dimensions beyond the current observation to also include recurrent network activations, so that an agent's internal thought vectors can themselves be stored as experiences for later recall, and can be used as read keys.
3. Rendering memory deletion a function of memory importance, so that certain experiences can be remembered longer than others.
4. Introducing an additional mask over dimensions for write operations, so that memories need not cover all available dimensions.

The human mind offers a remote, shining existence proof of general intelligence still beyond our reach. Despite the distance, it lights our path, and grows brighter with each step we take toward it.

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

# Appendices

## A    DNC EXPERIMENTS

The differentiable neural computer  (Graves et al., 2016) employs a differentiable memory matrix as external memory, which shares certain sequential addressing features with computer random access memory, with the objective of combining the advantages of neural and computational processing in one trainable system.

To explore why DNC never learned to use its memory matrix on the Concentration task, we applied DNC to a series of simpler tests of associative recall. The general pattern was that DNC performance seemed to deteriorate with episode length, as if DNC had difficulty reusing previously allocated locations in its memory matrix.

We found the simplest demonstration of this problem to be a small modification of the copy task included in DNC's GitHub repository. For Figures  4 and  5 the memory matrix was of size 16x16, exactly large enough to store the 16x16 random input data in each copy round, even if none of the data was stored in the LSTM (size 64). But when given two copy rounds in a sequence with no reset in between, the only way to achieve zero error was to reuse some locations in the memory matrix. This happened on two of our runs, but not on the other three runs. The instabilities in Figure 5 demonstrate the difficulty that DNC had in learning to reuse its memory matrix.

## B    MINI-SHRDLU

DNC (Graves et al., 2016) was evaluated on multiple supervised tasks and one RL task: Mini-SHRDLU. The Mini-SHRDLU task was actually composed of two separate sub-tasks: data buffering and puzzle solving. The constraints defining the problem, along with many other decoy constraints, were fed to the RL agent once while the agent was not allowed to work on the puzzle. Only after termination of the constraint presentation phase was the agent allowed to reposition the blocks to solve the puzzle. The results demonstrated that DNC used its external memory to buffer the incoming constraint information in the memory matrix, then used that data to achieve significantly better results on the combined buffer-puzzle task than did a baseline LSTM-based RL agent without external memory.

We considered using the Mini-SHRDLU task as a test of an RL agent's episodic memory. The data-buffering stage of the task did not seem relevant to this goal, since human memory seems ill-suited for memorizing long lists of data seen only once. Since the buffer and puzzle sub-tasks were not

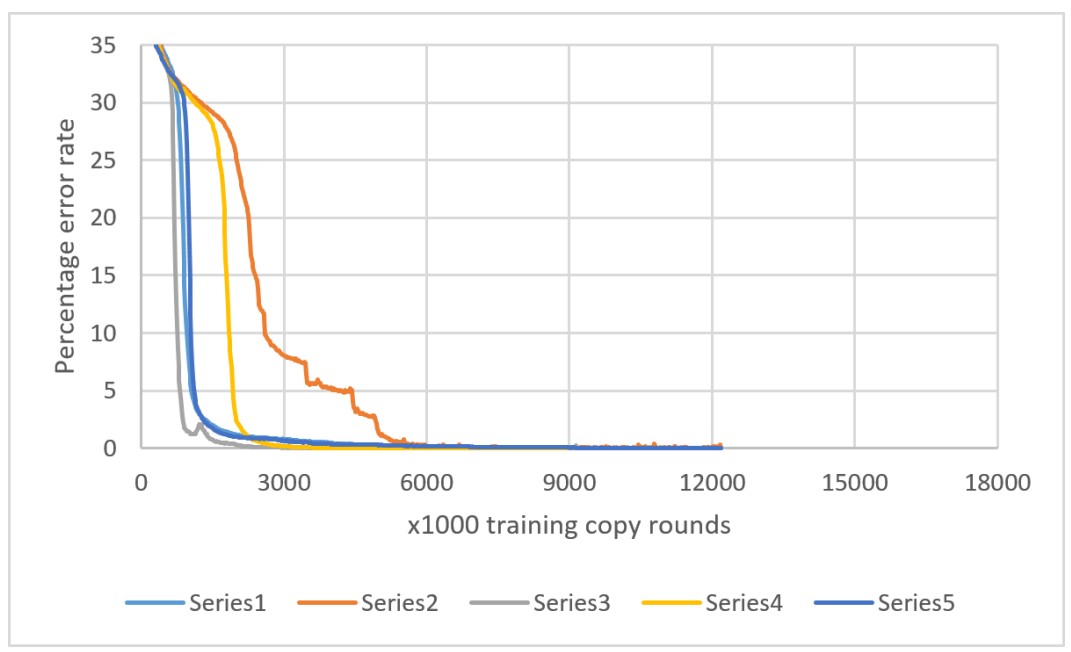

Figure 4: Multiple-copy task, 1 copy round. Five random runs where memory reuse is not required.

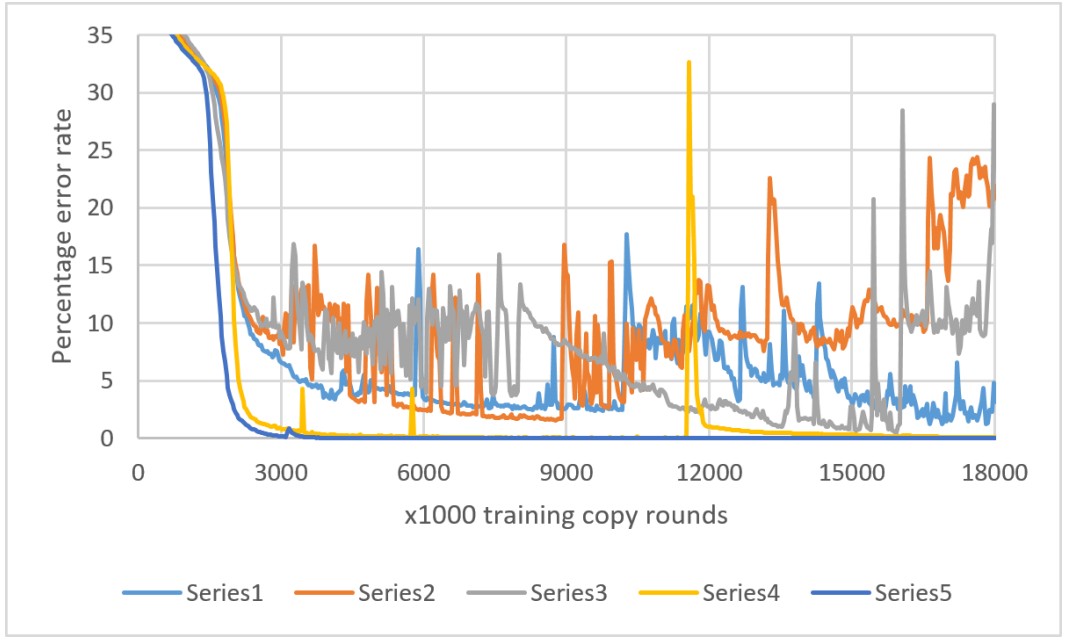

Figure 5: Multiple-copy task, 2 copy rounds. Five random runs where memory reuse is required.

evaluated separately, we couldn't be sure whether DNC's external memory helped in the puzzle-solving component of the combined task. We investigated this question by implementing Mini-SHRDLU without the data-buffering subtask, and giving an LSTM-based A3C RL agent access to a simple, non-differentiable, circular array of constraints. This allowed the agent to read the instructions at its own pace using 3 additional actions (move to next element, move to previous element, stay at current element).

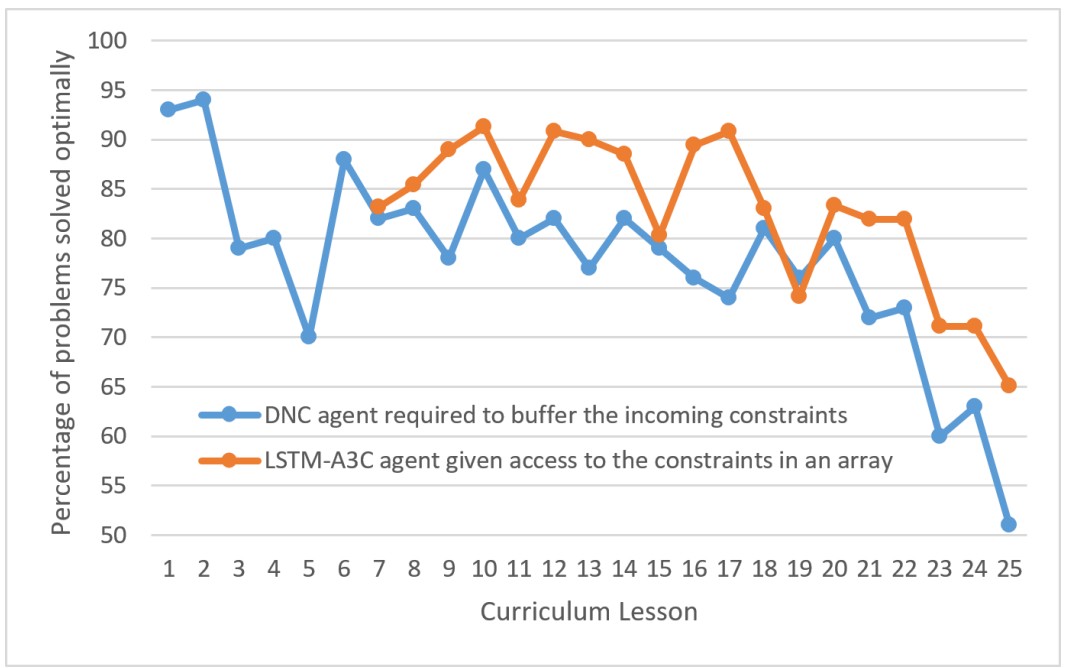

Figure 6: Mini-SHRDLU results. The DNC results were copied from (Graves et al., 2016) . The LSTM-A3C results were achieved by an RL agent with no external memory that read the constraints from a simple array at its own pace. After training on a lesson, the LSTM agent was tested on that lesson using 10,000 random problems. Direct numeric comparisons between these two performance curves are not meaningful for various reasons: (1) The LSTM agent was advanced to the next lesson only after its performance on the previous lesson had plateaued, which gave it more training problems than used by DNC. (2) The LSTM agent was always given problems using the full 6 blocks on the first 6 lessons, which is why those results are not shown. (3) Twelve of the subsequent lessons were actually skipped during training, and the LSTM agent was tested on those lessons using the model from the next trained lesson.

As shown in Figure 6, the LSTM-based RL agent learns to solve Mini-SHRDLU problems as well as DNC, but without using a differentiable memory matrix. These results demonstrate that external memory was not required for the problem-solving stage of the task.

