# OpenReview forum: "Now I Remember! Episodic Memory For Reinforcement Learning"
_ICLR.cc/2018/Conference — Reject_

### Official Review · AnonReviewer2 · 2017-11-26
**Interesting similarity function, but insufficient evidence of generality**

**Rating:** 4
**Confidence:** 5

**Review:**

There are a number of attempts to add episodic memory to RL agents. A common approach is to use some sort of recurrent model with a model-free agent. This work follows this approach using what could be considered a memory network with a identity embedding function and tests on 'Concentration', a game which requires matching pairs of cards. They find their model outperforms a DNC and LSTM baselines.

The primary novelty is the use of an explicitly masked similarity function (with learned mask) and the concentration task, which requires more memory than, for example, common tasks adapted from the psychology literature such as the Morris watermaze or T-maze (although in the supervised setting tasks such as Omniglot are quite similar).

This work is well-communicated and cites relevant prior work. The author's should also be commended for agreeing to release their code on publication.

The primary weakness of this work its lack of novelty and lack of evidence of generalization of the approach, which limits its significance. The model introduced is a slight variant of memory networks. Additionally, the single task the model is tested on appears custom-designed to favor the model (see next paragraph). While the analysis of the weakness of cosine similarity is interesting, memory networks which compute separate embeddings for the 'label' (content-based label for retrieval) and memory content don't appear to suffer from the same issue as the DNC. They can store only retrieval-relevant content in the label and thus avoid issues with normalization.

The observation vector is stored directly in memory without passing through an embedding function, which in general seems quite limiting. However, in the constructed task the labels are low-dimensional, random vectors and there is no noise in the labels (i.e. two cards with the same label are labelled identically, rather the similarly). The author's mention avoiding naturalistic labels such as omniglot characters (closer to the real version of concentration) due to the possibility the agent might memorise the finite set of labels, however by choosing a large dataset and using a non-overlapping set of examples for the test set this probably could be avoided and would provide a more naturalistic test set.

The comparison with the DNC also seems designed to favor their model. DNC has write-gates, which might be relevant in a task with many irrelevant observations, but in this task are clearly going to impair learning. A memory network seems the more appropriate comparison. Its not clear why the DNC model used two different DNCs for computing the policy and value.

To demonstrate their model is of more general interest it would be necessary to try on a wider range of more naturalistic tasks and a comparison with model-free agents augmented with memory networks. Simply showing that a customized model can outperform on a single custom, synthetic task is insufficient to demonstrate that these changes are of wider interest.

Minor issues:
- colorblind seems an odd description for agents which cannot perceive the card face. Why not just 'blind'? colorblind would seem to imply partial perception of the card face.

- the observations of the environment are defined explicitly, but not the action space.

---

### Official Review · AnonReviewer1 · 2017-11-26
**A modification of content-based memory retrieval and a new memory task**

**Rating:** 4
**Confidence:** 4

**Review:**

The paper addresses an important problem of how ML systems can learn episodic memory.
Authors, first, criticize the existing approaches and benchmarks for episodic memory, arguing that the latter do not necessarily test episodic memory to the full extent of human-level intelligence.
Then, a new external memory augmented network (MEM) is proposed which is similar in the spirit to content-based retrieval architectures such as DNC and memory networks, but allows to explicitly exclude certain dimensions of memory vectors from matching.
Authors evaluate the proposed MEM together with DNC and simple LSTM baselines on the game of Concentration where they find MEM to outperform concurrent approaches.

Unfortunately, I did not find enough of novelty, clarity or at least rigorous and interesting experiments in the paper to recommend acceptance.

Detailed comments:
1) When a new architecture is proposed, it is good to describe in detail, at least in the appendix. Currently, it is introduced only implicitly and a reader should infer the details from fig. 2.
2) It looks like the main difference between DNC and MEM is the way of addressing memories that allow explicit masking. If so, then to me this is a rather minor novelty and to justify it's importance authors should run a control experiment with the exact same architecture as in DNC, but with a masked similarity kernel. Besides that, an analysis of that is learned to be masked should be provided, how "hard" (i.e. strictly 0 and 1) are the masks, what influences them etc.
3) While the game of concentration clearly requires episodic memory to some extent, this only task is not enough for testing EM approaches, because there is always a risk that one of the evaluated systems somehow overfitted to this task by design. Especially to reason about human-level intelligence we need a variety of tasks.
4) To continue the previous point, humans would not perform well in the proposed task with random card labels, because it is very likely that familiar objects on cards help building associations and remembering them. Thus it is impossible to make a human baseline for this task and decide on how far are we below the human level.

---

### Official Review · AnonReviewer3 · 2017-11-27

**Rating:** 4
**Confidence:** 5

**Review:**

# Summary
This paper proposes an external memory architecture for dealing with partial observability. The proposed method is similar to Memory Q-Network [Oh et al.], but the paper proposes a masked Euclidean distance as a similarity measure for content-based memory retrieval. The results on "Concentration" task show that the proposed method outperforms DNC and LSTM.

[Pros]
- Presents a new memory-related task.

[Cons]
- No comparison to proper baselines and existing methods.
- Demonstrated in a single artificial task.

# Novelty and Significance
The proposed external memory architecture is very similar to MQN [Oh et al.]. The proposed masked Euclidean distance for similarity measure is quite straightforward. More crucially, there is no proper comparison between the proposed masked Euclidean distance and cosine similarity (MQN).

# Quality
- The paper does not compare their method against proper baselines (e.g., MQN or the same memory architecture with cosine similarity). DNC is quite a different architecture that has flexible writing/erasing with complex addressing mechanisms. Comparison to DNC does not show the effect of the proposed idea (masked Euclidean distance).
- The paper shows empirical results only on "Concentration" task, which is a bit artificial. In addition, the paper only shows a learning curve without any analysis of the learned model or qualitative results.

# Clarity
- Is the masked weight (w) a parameter or an activation of the network?
- The description of concentration task is a bit lengthy. It would be better to move some details to the appendix.
- I did not understand the paper's claim that "no existing RL benchmark task could unambiguously evaluate episodic memory in RL agents" and "In contrast, an episodic memory task like Concentration presents many previously unseen observations which must be handled correctly without prior exposure". In the Pattern Matching task from [Oh et al.], the agent is also required to compare two unseen visual patterns during evaluation.

---

### Decision · Program_Chairs · 2018-01-29
**ICLR 2018 Conference Acceptance Decision**

**Decision:**

Reject

**Comment:**

The authors show evidence that an RL agent with a new neural architecture with an external memory is superior on a version of the concentration game to a baseline.   However, other works have proposed neural architectures with episodic memories, and the reviewers feel that the proposed model was not adequately compared to these.  Furthermore, there are concerns about the novelty of the proposed model.